# Transcriptomic Differences Underlying the Activin-A Induced Large Osteoclast Formation in Both Healthy Control and Fibrodysplasia Ossificans Progressiva Osteoclasts

**DOI:** 10.3390/ijms24076822

**Published:** 2023-04-06

**Authors:** Ton Schoenmaker, Joy Zwaak, Bruno G. Loos, Richard Volckmann, Jan Koster, E. Marelise W. Eekhoff, Teun J. de Vries

**Affiliations:** 1Department of Periodontology, Academic Centre for Dentistry Amsterdam (ACTA), University of Amsterdam and Vrije Universiteit Amsterdam, 1081 LA Amsterdam, The Netherlands; 2Center for Experimental and Molecular Medicine, Amsterdam UMC Location University of Amsterdam, 1105 AZ Amsterdam, The Netherlands; 3Department of Internal Medicine Section Endocrinology, Amsterdam UMC Location Vrije Universiteit Amsterdam, 1081 HZ Amsterdam, The Netherlands; 4Rare Bone Disease Center Amsterdam, Bone Center, 1081 HV Amsterdam, The Netherlands

**Keywords:** fibrodysplasia ossificans progressiva, osteoclast, RNAseq, Activin-A, cell size

## Abstract

Fibrodysplasia Ossificans Progressiva (FOP) is a very rare genetic disease characterized by progressive heterotopic ossification (HO) of soft tissues, leading to immobility and premature death. FOP is caused by a mutation in the Activin receptor Type 1 (ACVR1) gene, resulting in altered responsiveness to Activin-A. We recently revealed that Activin-A induces fewer, but larger and more active, osteoclasts regardless of the presence of the mutated ACVR1 receptor. The underlying mechanism of Activin-A-induced changes in osteoclastogenesis at the gene expression level remains unknown. Transcriptomic changes induced by Activin-A during osteoclast formation from healthy controls and patient-derived CD14-positive monocytes were studied using RNA sequencing. CD14-positive monocytes from six FOP patients and six age- and sex-matched healthy controls were differentiated into osteoclasts in the absence or presence of Activin-A. RNA samples were isolated after 14 days of culturing and analyzed by RNA sequencing. Non-supervised principal component analysis (PCA) showed that samples from the same culture conditions (e.g., without or with Activin-A) tended to cluster, indicating that the variability induced by Activin-A treatment was larger than the variability between the control and FOP samples. RNA sequencing analysis revealed 1480 differentially expressed genes induced by Activin-A in healthy control and FOP osteoclasts with *p*(adj) < 0.01 and a Log2 fold change of ≥±2. Pathway and gene ontology enrichment analysis revealed several significantly enriched pathways for genes upregulated by Activin-A that could be linked to the differentiation or function of osteoclasts, cell fusion or inflammation. Our data showed that Activin-A has a substantial effect on gene expression during osteoclast formation and that this effect occurred regardless of the presence of the mutated ACVR1 receptor causing FOP.

## 1. Introduction

Osteoclasts are multinucleated cells that are specialized in the degradation of bone. They arise through the fusion of mononucleated monocyte-lineage-derived precursors under the influence of the receptor activator of the Nf-k ligand. A key question in osteoclast biology is what determines the ultimate size and number of nuclei of an osteoclast, and whether size is associated with activity Unexpectedly, we have recently shown that Activin-A, a Bone Morphogenetic Protein (BMP) family member that may trigger heterotopic ossification (HO) in fibrodysplasia ossificans progressiva (FOP), induces the formation of large osteoclasts, both in FOP osteoclasts and in healthy controls. Here, we explored the transcriptomic differences underlying the formation of this large osteoclast.

Fibrodysplasia ossificans progressiva is a severe, disabling autosomal-dominant genetic disorder [1], with a prevalence of 1 in 2 million cases worldwide. FOP is characterized by progressive heterotopic ossification, particularly in the ligaments, tendons and muscles [2,3]. The extraskeletal bone is formed via endochondral ossification [4]. Attachment of the new extraskeletal bones to the existing skeleton leads to cumulative immobility, a wheelchair-bound life and premature death.

In most FOP patients, the underlying cause is a single nucleotide c.617G > A mutation that substitutes arginine for histidine at Codon 206 (R206H of the BMP Type 1 receptor called Activin receptor Type 1/Activin Kinase 2 (ACVR1/ALK2) [2]. Research using induced pluripotent stem cells derived from patients with FOP, FOP cell lines and an FOP mouse model recently discovered that, in particular, Activin-A, a transforming growth factor (TGF)-β superfamily ligand [5], enhanced osteogenic differentiation in FOP via the mutated ACVR1^R206H^ receptor [6,7].

The focus of previous research into FOP has been on bone-forming cells, as the disease is characterized by extensive bone formation. The role of bone-degrading osteoclasts in FOP has not been extensively studied. However, normal bone remodeling and endochondral ossification require communication between the bone-forming osteoblasts and bone-degrading multinucleated osteoclasts [8,9]. This coupled action of osteoblasts and osteoclasts balances bone metabolism. Osteoclasts arise through the fusion of the mononuclear hematopoietic precursor stem cells located in the bone marrow and peripheral blood [10,11]. Osteoclast formation is mediated by the expression of Macrophage Colony Stimulating Factor (M-CSF) and Receptor Activator of Nuclear factor Kappa-B Ligand (RANK-L) by osteoblast-like cells, which, in turn, induces fusion of CD14-positive (CD14+) monocytic cells into multinucleated osteoclasts [8,12]. This multinucleation appears pivotal for the proper functioning of osteoclasts [13]. In FOP, there is unwanted extraskeletal bone formation that is not removed by the osteoclasts. Once formed, these bones undergo normal bone remodeling, implying that at that stage, the coupled action of osteoblasts and osteoclasts has normalized. The role of the ACVR1^R206H^ mutation on the formation of osteoclasts is unclear, as is the effect of Activin-A on the formation of human osteoclasts in general. In a first report on the role of Activin-A in the formation of osteoclasts, we recently described that Activin-A induces fewer, but larger and more active, osteoclasts regardless of the presence of the ACVR1^R206H^ mutation in the osteoclasts’ precursors [14]. We described that the osteoclasts that were formed in the presence of Activin-A were at least fourfold larger. Osteoclasts thus formed were more active in bone resorption and expressed more Cathepsin K, one of the crucial proteases for bone resorption. This implies an increased fusion of the osteoclast precursors or even of the multinucleated osteoclasts with existing multinucleated osteoclasts. However, the underlying molecular mechanisms of this effect of Activin-A remains unknown. In the present study, we used RNA sequencing as an open-ended, non-biased approach to explore the pathways and genes associated with this process. Samples were taken after 14 days of culture, which was in the dynamic phase, during which the initial formation of multinucleated osteoclasts in these kinds of cultures normally occurs. We also showed that at the gene expression level, differences between control and FOP cells were neglectable. However, Activin-A induced major transcriptomic changes in both the control and FOP cells during osteoclastogenesis. This provided a new fundamental insight into the molecular effect of Activin-A on the formation of human osteoclasts.

## 2. Results

### 2.1. Experimental Treatment Samples Clustered Together

After RNA quality control, one control and one FOP sample were excluded from further analyses because they showed a RNA Inrgerity Value (RIN) value of < 7 (Control 3 + Activin-A and FOP 2 − Activin-A (Appendix A).

Further analyses was performed on 11 control samples (six samples without Activin-A and five samples with Activin-A) and 11 FOP samples (five samples without and six samples + Activin-A).

We first performed principal component analysis (PCA) to assess the inter- and intragroup variability. Figure 1A depicts the inter-donor variability between the samples, showing no clear clustering. In Figure 1B, on the other hand, clear intra-group clustering can be seen between the experimental conditions (without Activin-A and with Activin-A) in both the control and FOP samples. This clustering was also apparent in the Pearson’s correlation heatmap visualizing the correlation (r) values between the samples (Figure 1C), showing that the addition of Activin-A and not the FOP genotype was the main driving factor for the clustering. There was no such strong clustering based on sex (Appendix A).Taken together, these findings show that the variability induced by the Activin-A treatment was higher than the variability among the different donors, and that the control and FOP samples reacted in a similar way to the presence of Activin-A.

### 2.2. Activin-A Activates Osteoclast-Related Pathways

Since the control and FOP samples reacted in a similar way to Activin-A, and the samples from the experimental groups clustered together independently of the presence of the mutated ACVR1 receptor (Figure 1B), we grouped both the control and FOP samples with the same experimental conditions together for further analysis (e.g., control and FOP without Activin-A vs. control and FOP +Activin-A).

When applying cut-off values of *p*(adj) < 0.01 and Log2 fold change ≥2 to the expression data to identify differentially expressed genes (DEGs), we found 1480 significantly differentially expressed genes in the presence of Activin-A, 566 of which were upregulated and 914 were downregulated. The top 100 most significant DEGs were divided into upregulated and downregulated genes, and are shown in Appendix A. We found an association with several Kyoto Encyclopedia of Genes and Genomes (KEGG) pathways for the genes that were upregulated in the presence of Activin-A but, surprisingly, none for the downregulated genes (Figure 2). Genes upregulated by Activin-A showed an association with different osteoclast differentiation or function pathways involved in, among others, hematopoietic lineage, cell adhesion molecules (CAMs), PI3K/Akt signaling, Th17 cell differentiation and Rap1 signaling pathways. This suggests that Activin-A is associated with signaling pathways involved in the differentiation and function of osteoclasts.

We performed Gene Ontology (GO) analysis to identify the biological processes, molecular functions and cellular components associated with the differentially expressed genes regulated by Activin-A. Bubble plots of the top 20 GO terms of each of the three GO domains that were associated with the genes upregulated in the presence of Activin-A are depicted in Figure 3A–C. Bubble plots of the top 20 GO terms most significantly associated with the downregulated genes are depicted in Figure 4A–C. For the upregulated genes, more than 900 GO terms were over-represented and for the downregulated genes more than 350 GO terms were over-represented in the biological process domain. Some of these GO terms that were most significantly associated with the upregulated genes could be linked to the process of osteoclast differentiation such as “regulation of cell differentiation” (GO:0045595), “extracellular matrix organization” (GO:0030198), “positive regulation of cell population proliferation” (GO:0008284), “cell migration” (GO:0016477), “chemotaxis” (GO: 0006935), “cell adhesion” (GO:0007155) and ‘‘inflammatory response (GO: 0006954) (Figure 3B).

Several GO terms were significantly associated with the downregulated genes and could be related to osteoclast formation and Activin-A signaling, such as “cell morphogenesis involved in differentiation” (GO:0000904), “negative regulation of ERK1 and ERK2 cascade” (GO:0070373), “negative regulation of cell migration” (GO:0030336), “negative regulation of cell adhesion” (GO:0007162), “negative regulation of transforming growth factor beta receptor signaling pathway” (GO:0030512) and “ossification” (GO:0001503).

We next focused on the biological processes in more detail. Cell fusion, such as that seen in osteoclast formation, can be roughly divided into five distinct processes, namely differentiation, actin reorganization, cell migration, cell adhesion and membrane fusion [15,16]. Interestingly, of all the GO terms in the biological process domain associated with the upregulated genes, several could be clearly divided according to these different processes, as depicted in the bubble plot in Figure 5A.

### 2.3. Activin-A-Induced Differentially Expressed Genes Can Be Linked to Cell Fusion

To identify which specific genes were differentially expressed under the influence of Activin-A, we next focused on the expression of individual genes in the different groups. The heatmap in Figure 6A shows the clear clustering of the top 100 DEGs between samples without and with Activin-A. Moreover, the volcano plot (Figure 6B) shows a clear pattern of upregulated and downregulated genes under the influence of Activin-A. The top 100 most significant DEGs were divided into the upregulated and downregulated genes, and are shown in Appendix A. To confirm the RNAseq results, the expression of some of these genes was also analyzed by qPCR (Figure 7 and Figure 8). These genes were selected on the basis of the volcano plot (Figure 6); we selected the most distinctly upregulated or downregulated genes.

Again, when we explored the cell fusion process further, 20 of the upregulated genes could be linked to differentiation, 54 to the organization of actin, 26 to migration/chemotaxis, 18 to adhesion and 36 to processes linked to membrane fusion (Appendix A). Several of these genes have already been associated with cell differentiation (*HTRA1* and *CIITA* [17,18]), actin organization (*EPHA2* and *RHOBT1B* [19,20]), cell adhesion and migration (*CXCL12* and *S1PR1* [21,22]), and membrane fusion (*DNM1* [23]), but others have not previously been associated with the formation of osteoclasts.

Some of the genes that showed the most distinct upregulation, as shown in the volcano plot and confirmed to be overexpressed by qPCR (Figure 7), showed interesting associations with different GO terms, making these genes interesting candidates for further research on the mechanisms underlying the formation of large osteoclasts. Some of these genes are described further in the discussion. The downregulated genes (Figure 8) did not show such a clear association with cell fusion processes, although *NEDD9* has been shown to be downregulated during RANK-L-induced osteoclast differentiation [24].

## 3. Discussion

The present study identified a multitude of mRNAs that are regulated during the formation of large osteoclasts, such as those induced by Activin-A. The study was performed both on osteoclasts formed from monocytes from FOP patients with a mutation in ACVR1 and from controls without this mutation. In both cases, large osteoclasts were observed, indicating that the mechanism by which large osteoclasts are formed, is comparable, corroborated by the findings that regulation by Activin-A was similar between the controls and FOP patients. In order to gain mechanistical insights, we performed RNA sequencing analysis on these cultures.

It has been described that Activin-A sends signals via the SMAD2/3 pathway and normally inhibits signaling via ACVR1 and subsequent SMAD1/5/8 phosphorylation by forming a non-signaling complex (NSC) with this receptor [25]. The mutation in the ACVR1 gene in FOP is believed to alter the sensitivity of the receptor and to result in Activin-A stimulation via ACVR1 rather than inhibition [6,7]. The fact that we did not observe any differences in the response to Activin-A between the control and FOP cells with regard to the formation of osteoclasts suggests that this signal is not mediated via the mutated ACVR1 receptor but via the canonical SMAD2/3 signaling pathway, a pathway that was previously identified in osteoclasts [26]. Our findings here, with a lack of an FOP mutation specifically induced by Activin-A, is in contrast to our previous study, where we showed that Activin-A exclusively regulated genes present in the FOP mutation containing periodontal ligament fibroblasts [27]. Apparent cell-type-specific differences in the concentrations and combinations of Type I and Type II bone morphogenic protein receptors between periodontal ligament fibroblasts, and osteoclasts and their precursors could explain such a different response for Activin-A between these cell types [28,29,30]. This lack of a disease-specific effect of Activin-A on osteoclasts could suggest that other Activin-A receptors on the osteoclast precursors overruled the mode of action of Activin-A in binding to ACVR1, albeit that ACVR1 is transcribed in osteoclasts and that the mutated form was transcribed at a ninefold higher level compared with the non-mutated form in osteoclasts derived from FOP patients [14]. ACVR1B, ACVR2A and ACVR2B were not regulated by Activin-A, whereas ACVR1C was downregulated.

The genes differentially expressed in the presence of Activin-A showed a clear association with biological processes that are linked to cell fusion. In particular, the upregulated genes could be divided into the different stages of cell fusion. Not surprisingly, some of these genes showed some overlap and were associated with several stages of cell fusion. One of these molecules is OLR1, also known as LOX1, which showed an association with several biological processes involved in cell fusion, such as GO:0016477 “cell migration”, GO:0099500 “vesicle fusion to plasma membrane”, GO:0098609 “cell–cell adhesion” and GO:0061025 “membrane fusion”, suggesting an important role for this molecule in the Activin-A enhanced fusion process. This receptor has been described as playing a role in the formation of osteoclasts, but with conflicting results, probably dependent on its location and local activator. *OLR1* activated by NF-κB plays an enhancing role in adhesion and migration in monocytic and breast cancer cells [31]. However, when activated by oxidized *LDL*, *OLR1* inhibits the migration of macrophages [32]. Nakayatchi et al. showed that *OLR1* reduces cell–cell fusion and thereby acts as a negative regulator of the formation of osteoclasts, although no effect on osteoclast genes such as *NFATc1* and *Cathepsin K* was observed [33,34]. On the other hand, when *OLR1* is activated, it stimulates the expression of adhesion molecules and pro-inflammatory signaling pathways and proangiogenic proteins in macrophages such as NF-kB and VEGF [35]. It has been proposed that VEGF itself may replace M-CSF during osteoclastogenesis [36]. Another possible explanation for the large osteoclasts that are formed in the presence of Activin-A could be found in the upregulation of *RHOBTB1*, which is associated with GO:0051493 “regulation of cytoskeleton organization”. This RhoGTPAse family member has been shown to be downregulated by *OPG*, resulting in decreased osteoclast activity due to disrupted formation of the sealing zone [37]. Interestingly, RhoGTPAses have been suggested to play a role in the self-contact-induced membrane fusion observed in epithelial cells, which is a very rapid and efficient way of cell fusion [38]. In short, epithelial cells develop actin-rich protrusions, eliminating the need for cell junctions for membrane fusion [38]. This process may also play a role when the membranes of two (pre-)osteoclasts are in contact, resulting in a rapid and efficient way of cell fusion called contact-induced fusion. In terms of cell fusion genes, it is remarkable that the strong candidates for cell fusion, *DC-STAMP* and *OC-STAMP*, were not regulated by Activin-A at the time of investigation. Possibly, the impact of high or even low expression levels of *DC-STAMP* in fusogenic cells [39] occurs at an earlier time point. In our previous study, we also showed a downregulation of *DC-STAMP* on Day 7, whereas no differences in expression were found at later timepoints.

There are only a few cell types in our body that are multinucleated and arise through the fusion of their precursors. Next to osteoclastogenesis, another process were cell fusion is essential is the formation of multinucleated myofibers by the fusion of myoblasts. Interestingly, some of the genes that were differentially expressed in this experiment have also been described as playing a role in myoblast fusion. Dynamin-1 (*DNM1*) was one of the upregulated genes associated with GO:0061025 “membrane fusion” and has previously been associated with osteoclast precursors and myoblast fusion [23]. The upregulated genes ADAM metallopeptidase domain 12 (*ADAM12*), Actinin Alpha 2 (*ACTN2*) and adhesion G protein-coupled receptor B1 (*ADGRB1*) were all associated with GO:0061025 “membrane fusion” and have been previously described as playing a role in myoblast fusion [40,41]. *ADAM12* was shown to be present in later stages of the formation of human osteoclasts [42] and to promote myoblast fusion via *ACTN2* binding [43]. *ACTN2* was previously found to be expressed in osteoclasts [43,44]. Since HO formation often starts at the site of injured muscles, probably partly mediated by locally increased levels of Activin-A due to an inflammatory response as a result of the damage [45,46], it is worthwhile to explore the different and similar genes and processes involved in the fusion of both myoblasts and monocytes further.

Activin-A has been described as an inflammation marker [47,48], and one of the molecules we found to be associated with GO:006954 “inflammatory response” was *CIITA*, a molecular switch of antigen presentation that is essential for the expression of genes encoding major histocompatibility complex (MHC)-II molecules. Benascuitti et al. showed that the overexpression of *CIITA* in monocytes enhanced osteoclastogenesis mediated by increased c-fms and RANK signaling [18], a process that, in our experiments, could also induce enhanced fusion. Interestingly, *CIITA* has also been shown to be involved in regulating MHC Class II molecules, thereby playing a role in antigen presentation and immune responses. The regulation of *CIITA* expression is complex and appears to be location- and cell-type-specific [49]. It has been proposed that osteoclasts, alongside their physiological role as bone-resorbing cells, also have a more immunological function as antigen-presenting cells and can play a role in the immune response [50,51]. Possibly, the Activin-A-induced upregulation of *CIITA* shown here steers the osteoclasts towards a more immunological cell under local high concentrations of Activin-A.

Although normal bone modeling takes place in heterotopic bone, it cannot be excluded that the osteoclasts in heterotopic bone are different from those in normal bones. Although human osteoclasts in vitro are cultured from blood-borne monocytes worldwide, it is a shortcoming of such an approach that the microenvironment of (heterotopic) bone, which could contain more relevant osteoclast precursors, cannot be accounted for. Another limitation of the approach used here is the fact that only one timepoint was investigated. Osteoclast formation is a dynamic biological process with different stages, during which specific genes play a role. Since we were interested in the mechanism underlying the formation of especially large osteoclasts, we chose a timepoint at which we had previously seen that the real process of fusion was underway rather than an earlier timepoint were we normally do not see any fusion taking place. Moreover, this is a descriptive study, where we explored the transcriptomic differences underlying the formation of large osteoclasts.

Future experimental validation of the exact role of the described genes will be performed to explore this biological process further. The validation we performed was a qPCR analysis of the most highly influenced genes from the same samples as used in the RNAseq itself. It would be worthwhile to also perform this validation on a separate set of RNA samples. It is, however, very difficult to draw blood from FOP patients; therefore, we were limited to the current experimental setup.

Taken together, the results of the present study showed that Activin-A induced gene expression that can be linked to the observed phenotype of larger and more active osteoclasts, which were described in our previous study, independent of the mutated ACVR1. The transcriptomic changes induced by Activin-A were similar in the control and FOP cells. In our earlier study, we showed the formation of large osteoclasts and the upregulation of osteoclast-specific genes at later timepoints under the influence of Activin-A. Here, we confirmed, on the gene expression level, that there was no difference in the effect of Activin-A on the formation of osteoclasts between the control and FOP cells. We identified genes that could be possibly involved in the formation of these large osteoclasts. Among the general osteoclast biology terms, our results identified Activin-A upregulated genes that could be associated with enhanced fusion during the formation of osteoclasts. Together with the downregulated genes that are associated with the negative regulation of cell adhesion and migration, this may explain the formation of the larger and more actively resorbing osteoclasts in the presence of Activin-A [14]. One of the potential modalities to reduce the formation of HO in FOP is to neutralize Activin-A with specific antibodies. This report showed that this could impact osteoclasts’ biology in general, possibly resulting in less active and smaller osteoclasts, justifying further investigation into the effect on the function of osteoclasts in general and in FOP.

## 4. Material and Methods

### 4.1. Osteoclastogenesis and RNA Isolation

The RNA samples used for the RNA sequencing and subsequent qPCR analyses were obtained from the osteoclastogenesis experiment described previously [14]. Briefly, blood was drawn from six sex- and age-matched controls and patients (2 males, 4 females; age range: 20–68 years; maximal age difference between the controls and FOP patients: 4 years; Table 1). Five of the FOP patients harbored the R206H mutation, and one patient harbored a variant mutation (Q207E). This Q207E mutation is adjacent to the classical R206H mutation and is also located in the GS domain. Patients with this mutation exhibited similar phenotypes to the R206H patients [52,53]. Written informed consent was obtained from all donors in accordance with the requirement of the Medical Ethics Review Committee of the Amsterdam UMC, Vrije Universiteit Amsterdam (research protocol 2012.467). CD14+ monocytes were isolated using CD14-antibody tagged microbeads (Miltenyi Biotec, Bergisch Gladbach, Germany). Cells were cultured in a 96-well plate at a density of 1 × 10^5^ cells/well for the first 3 days with 25 ng/mL of macrophage colony-stimulating factor (M-CSF) (R&D systems, Oxon, UK), without or with 50 ng/mL of Activin-A (Sigma, St. Louis, MO, USA). After 3 days, the composition of the medium was changed to 10 ng/mL of M-CSF and 2 ng/mL of receptor activator of nuclear factor kappa-B ligand (RANKL) (R&D Systems), without or with 50 ng/mL of Activin-A. The RNA was isolated after 14 days using the RNeasy mini kit (QIAGEN, Hilden, Germany) following the manufacturer’s instructions.

### 4.2. RNA Sequencing

Preceding the actual RNA-seq analysis, the RNA’s quality was assessed at the Core Facility Genomics of Amsterdam UMC using high-sensitivity RNA ScreenTape analysis (Agilent Technologies, Santa Clara, CA, USA) on the Agilent 2200 TapeStation system, following the manufacturer’s instructions. Samples not meeting the requirement of a RNA integrity number equivalent (RINe) value of ≥7 were excluded from further analyses [54].

Preparation of the samples and library was performed using the Kapa mRNA HyperPrep kit (Roche, Basel, Switzerland). Single-end 50 bp RNA sequencing was performed on the Illumina HiSeq 4000 platform (Illumina, San Diego, CA, USA). The reads were aligned to the human reference genome GRCh38 using HiSat version 2.2.1. Subsequently, the alignments were processed with the Count feature (Version 2.0.1) to produce a count matrix. The steps of library preparation, sequencing and data processing were performed by the Core Facility Genomics of Amsterdam UMC.

### 4.3. Real-Time Quantitative Polymerase Chain Reaction (qPCR)

After RNA isolation using the RNEasy mini kit, cDNA synthesis was performed using the first-strand cDNA synthesis kit (Thermo Fisher Scientific, Waltham, MA, USA) according to the manufacturers’ protocol, using both the Oligo(dT)18 and the D(N)6 primers. The qPCR primers were designed using Primer Express software, version 2.0 (Applied Biosystems, Foster City, CA, USA) (Table 2). To avoid the amplification of genomic DNA, each amplicon spanned at least one intron. The qPCR was performed on an LC480 light cycler (Roche, Basel, Switzerland). For this, 3 ng cDNA was used in a total volume of 20 µL containing the Light Cycler SybrGreen1 Master Mix (Roche) and 1 µM of each primer. A standard two-step qPCR program with an annealing temperature of 63 °C was performed. The expression of the housekeeping gene porphobilinogen deaminase (*PBGD*) was not affected by the experimental conditions. Samples were normalized to the expression of PBGD by calculating the ΔCt (Ct of the gene of interest — Ct of *PBGD*) and the expression of the different genes was expressed as the 2-(ΔCt) value. All qPCRs had equal efficiencies.

### 4.4. Data Analysis and Statistics

Inter- and intragroup variability were assessed and visualized with non-supervised principal component analysis (PCA) plots combined with Pearson’s correlation heatmaps visualizing the correlation (r) values between the samples. Differentially expressed genes (DEGs) were identified using the DESeq2 package with the standard settings on the R2: Genomics Analysis and Visualization Platform (http://r2.amc.nl). Volcano plots were generated using the same platform. Correction for multiple testing was performed using a false discovery rate (FDR) threshold, which was adaptive to the amount of signal in our data. An adjusted *p*-value (*p*(adj)-value) of <0.01 was considered to be the critical value to determine significance.

Pathway and gene enrichment analyses were performed on the differentially expressed genes with *p*(adj)-value < 0.01 and Log2 fold change ≥±2. Both the pathway analysis and the gene enrichment analysis were performed with ToppGene suite application ToppFun (March 2021 version) [55] using the standard settings.

For the qPCR data, systematic differences between related samples without Activin-A and with Activin-A were statistically tested using the Wilcoxon matched pairs signed rank test. The Mann–Whitney test was used to determine the systematic differences between the independent FOP and control samples. Differences were considered to be significant when *p* < 0.05. 

## Figures and Tables

**Figure 1 ijms-24-06822-f001:**
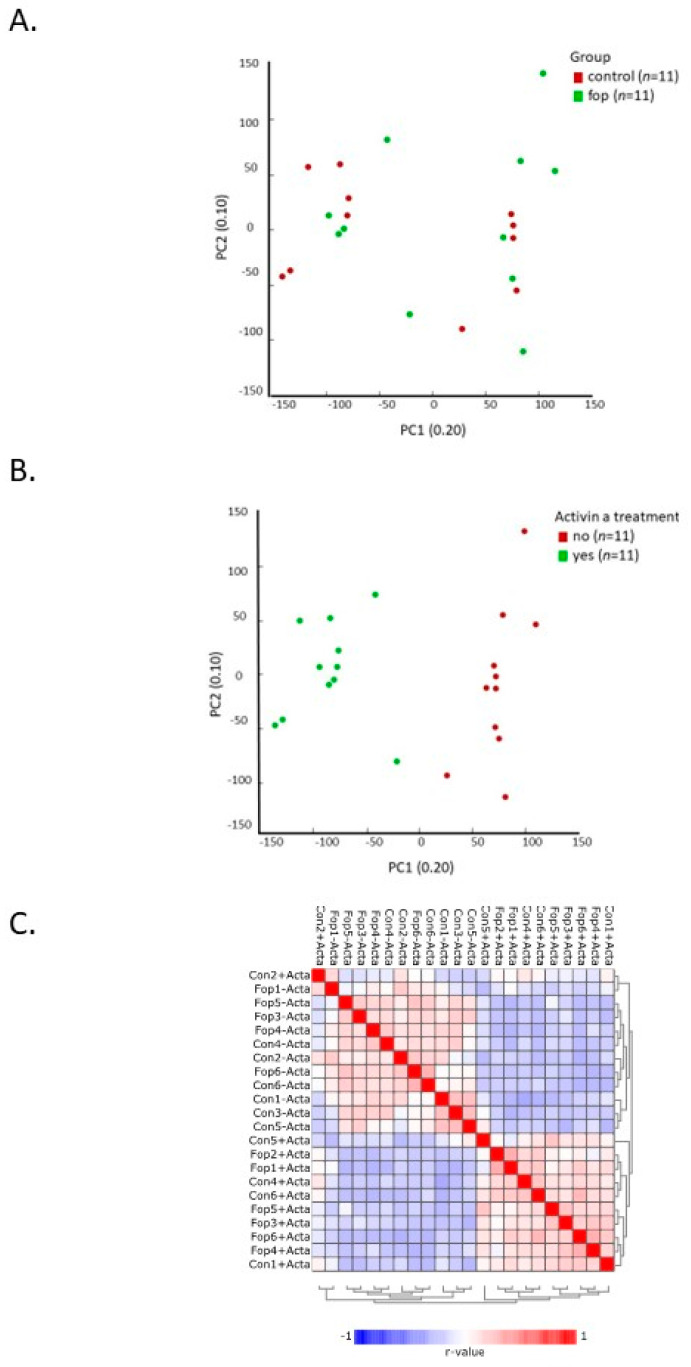
Samples cluster by experimental conditions. PCA plots and heatmap showing clustering of the samples. (**A**) PCA plot depicting inter-donor variability. −Activin-A and +Activin-A samples were grouped together per donor group. (**B**) PCA plot depicting inter-experimental variability. Control and FOP samples per experimental group (e.g., −Activin-A and +Activin-A) were grouped together. (**C**) Pearson’s correlation heatmap showing the grouping of samples with similar experimental conditions. Both the non-supervised principal component analysis and the Pearson’s correlation showed that samples with the same experimental treatment clustered together, indicating that the variation induced by the treatment was larger than the variation induced by the different donors (control *n* = 11 in total (without and with Activin-A); FOP *n* = 11 in total (without and with Activin-A)).

**Figure 2 ijms-24-06822-f002:**
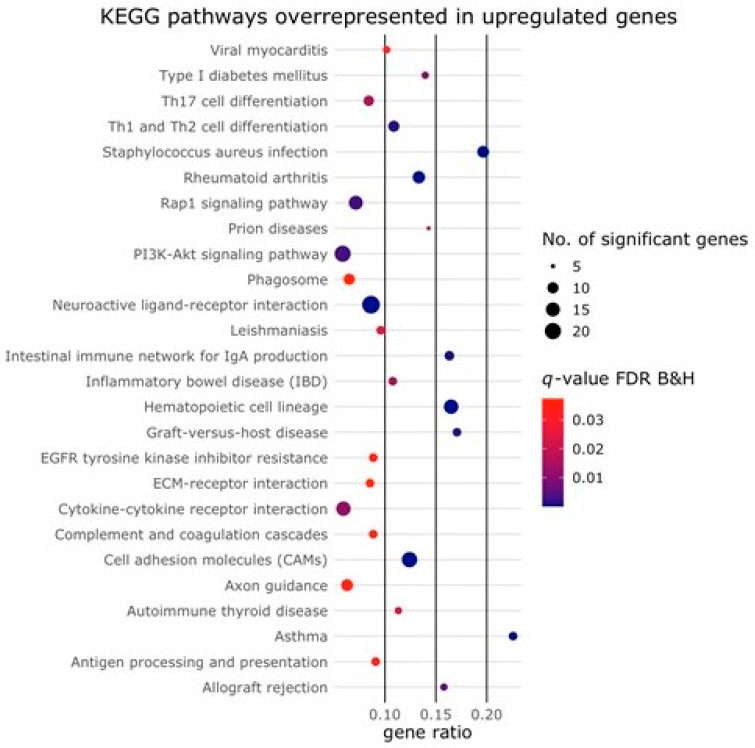
KEGG pathways enriched among the upregulated genes. Bubble plot showing KEGG pathways enriched among the genes upregulated by the Activin-A treatment. No enrichment was found among the downregulated genes. The size of the circle represents the number of DEGs (*p*(adj) < 0.01, Log2 fold change ≥2) in that pathway. The color of the circle represents the significance depicted as a q-value, with default settings for the Benjamini–Hochberg false discovery rate (FDR B&H). The gene ratio represents the ratio between the number of DEGs in our dataset and the total number of genes in that pathway.

**Figure 3 ijms-24-06822-f003:**
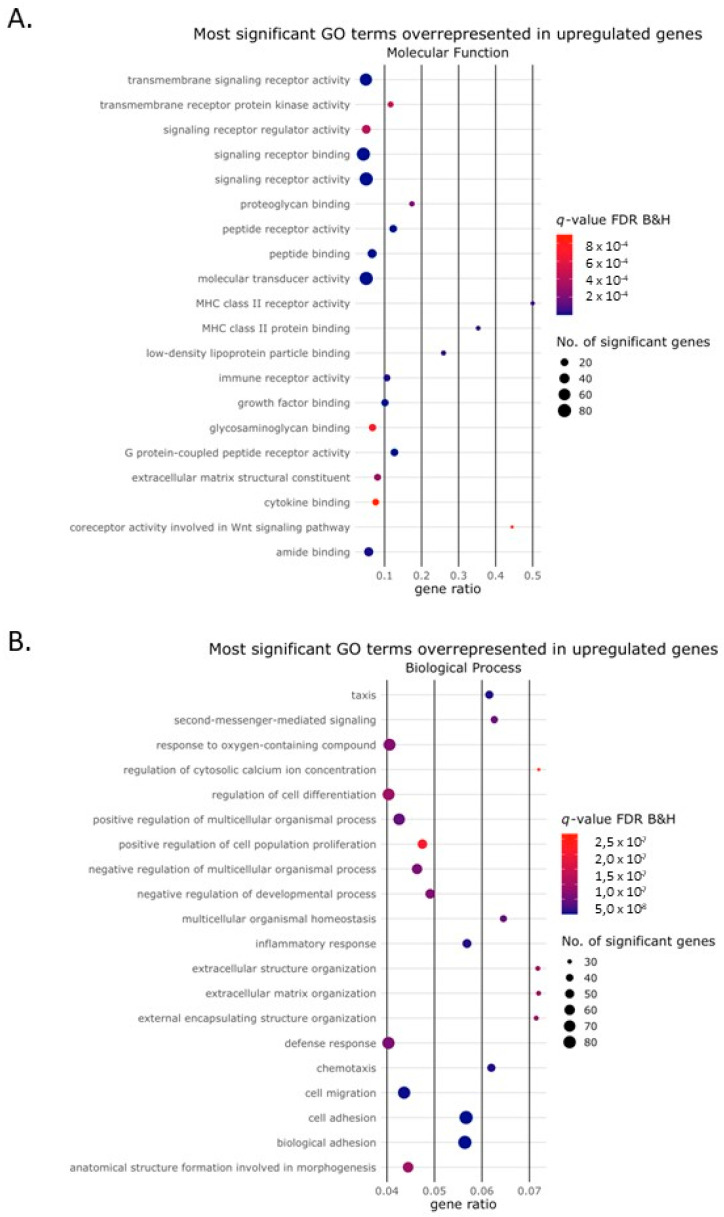
Gene Ontology terms enriched among the upregulated genes. Bubble plot showing Gene Ontology (GO) terms that were enriched among the genes upregulated by the Activin-A treatment. The size of the circle represents the number of DEGs (*p*(adj) < 0.01, Log2 fold change ≥2) in that pathway. The color of the circle represents the significance depicted as the q-value, with the default settings of the Benjamini–Hochberg false discovery rate (FDR B&H). The gene ratio represents the ratio between the number of DEGs in our dataset and the total number of genes in that GO category. The top 20 most significant GO terms were enriched in (**A**) the molecular function domain, (**B**) the biological process domain and (**C**) the cellular component domain.

**Figure 4 ijms-24-06822-f004:**
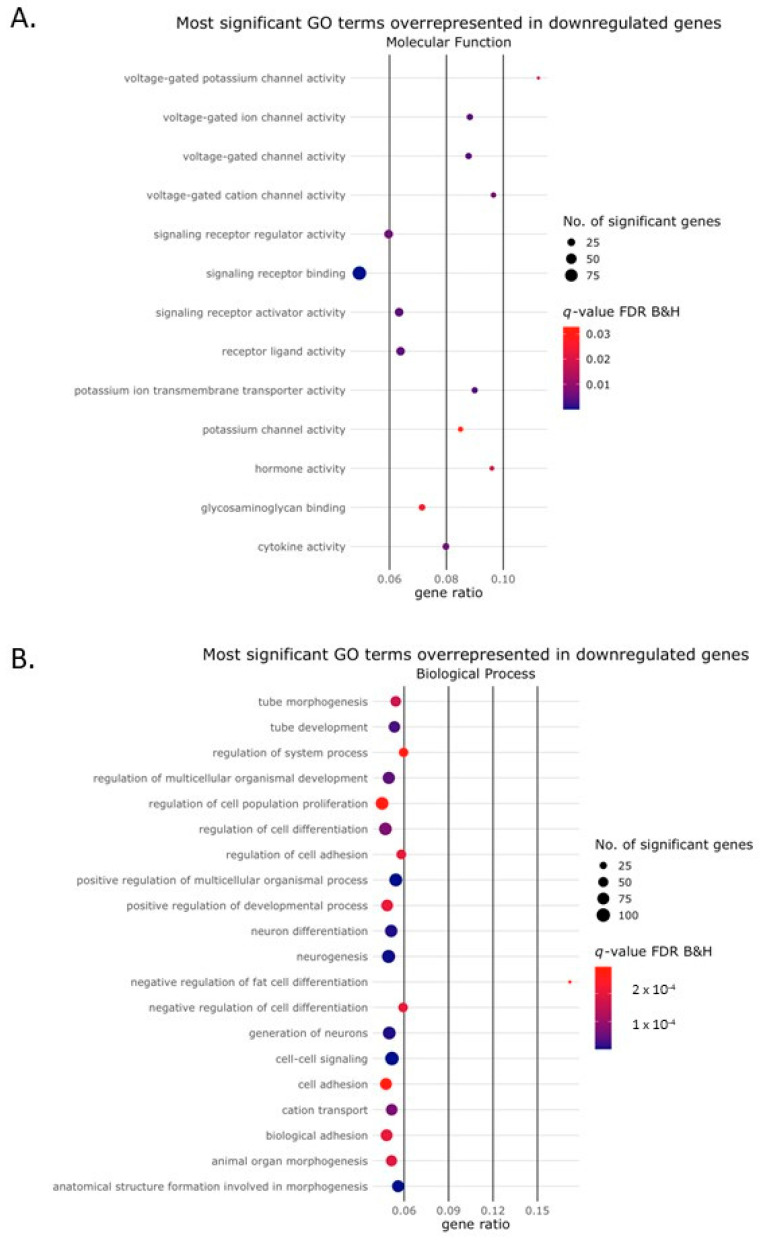
Gene Ontology terms enriched among the downregulated genes. Bubble plot showing the Gene Ontology (GO) terms that were enriched among the genes downregulated by the Activin-A treatment. The size of the circle represents the number of DEGs (*p*(adj) < 0.01, Log2 fold change ≥2) in that pathway. The color of the circle represents the significance depicted as a q-value, with the default settings of the Benjamini–Hochberg false discovery rate (FDR B&H). The gene ratio represents the ratio between the number of DEGs in our dataset and the total number of genes in that GO category. The top 20 most significant GO terms were enriched in (**A**) the molecular function domain (only 13 GO terms in this domain were found to be enriched), (**B**) the biological process domain and (**C**) the cellular component domain.

**Figure 5 ijms-24-06822-f005:**
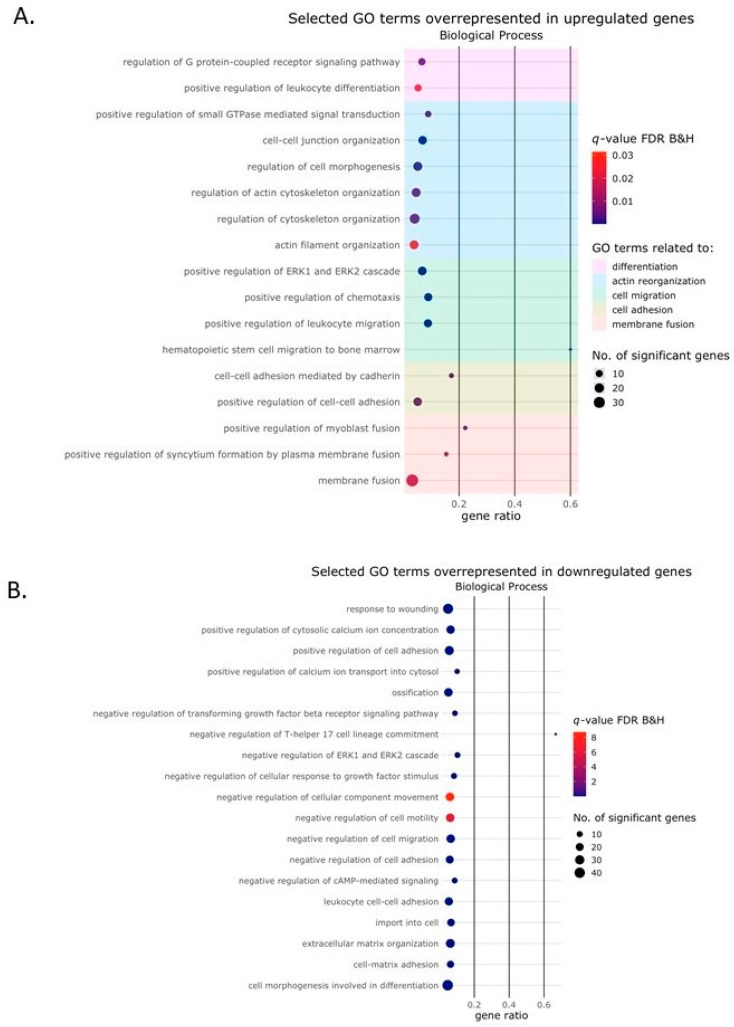
Gene ontology terms in the biological process domain. Bubble plot showing selected Gene Ontology (GO) terms in the biological process domain enriched among the genes that were upregulated or downregulated by the Activin-A treatment. The size of the circle represents the number of DEGs (*p*(adj) < 0.01, Log2 fold change ≥2) in that pathway. The color of the circle represents the significance depicted as a q-value, with the default settings of the Benjamini–Hochberg false discovery rate (FDR B&H). (**A**) Selected GO terms were enriched among the upregulated genes could be divided into the different biological processes involved in cell fusion. The different stages of the cell fusion process are depicted in different colors (**B**) Selected GO terms enriched among the downregulated genes that could be linked to osteoclasts in general.

**Figure 6 ijms-24-06822-f006:**
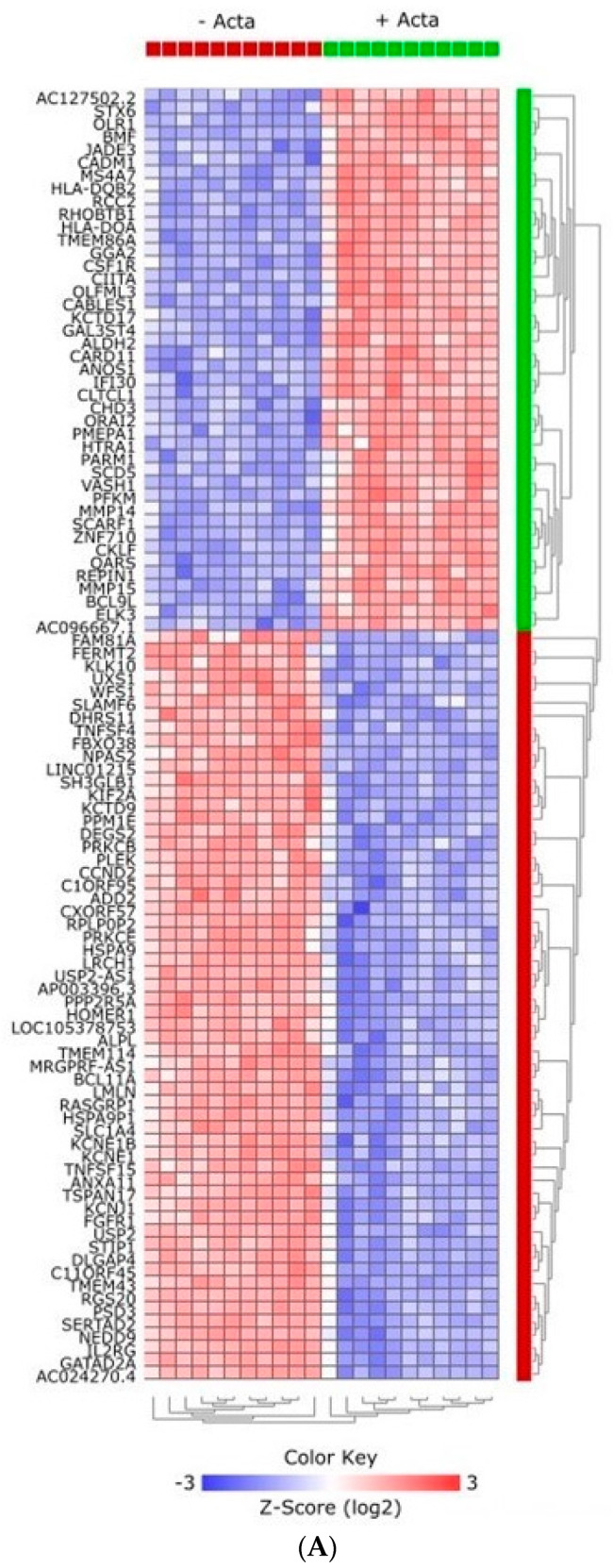
Activin-A-induced differential gene expression. Activin-A-induced DEGs with *p*(adj) < 0.01, Log2 fold change ≥2. (**A**) Heatmap showing the 100 most significant DEGs under the influence of Activin-A. (**B**) Volcano plot showing the upregulated and downregulated genes under the influence of Activin-A, plotted as the Log2 fold change (*x*-axis) versus the statistical significance (*y*-axis). Annotated dots are the genes confirmed by qPCR (see Figure 7).

**Figure 7 ijms-24-06822-f007:**
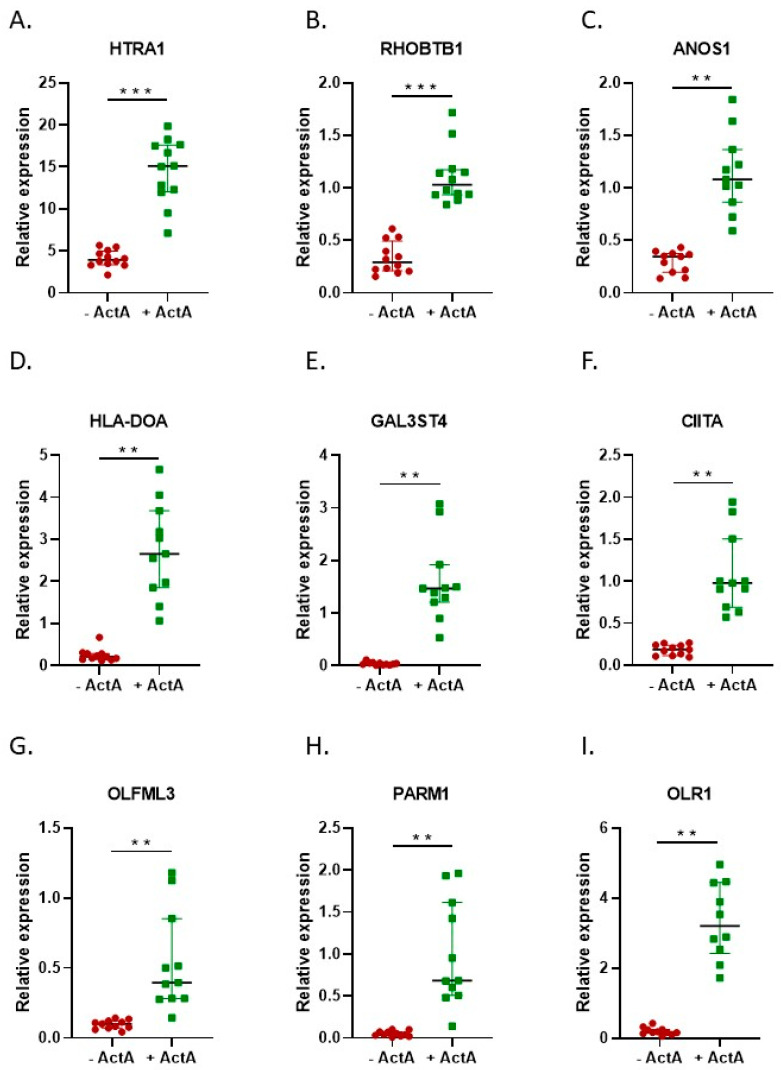
The qPCR confirmed the RNA-seq data. qPCR data for the most significantly differentially expressed genes depicted in the volcano plot. Data are presented as medians with the interquartile range. The qPCR data confirmed the RNA sequencing data. (**A**–**I**) qPCR of genes upregulated by Activin-A. −Activin-A, *n* = 11; +Activin-A, *n* = 11. Wilcoxon matched pairs signed rank test ** *p* < 0.01, *** *p* < 0.001.

**Figure 8 ijms-24-06822-f008:**
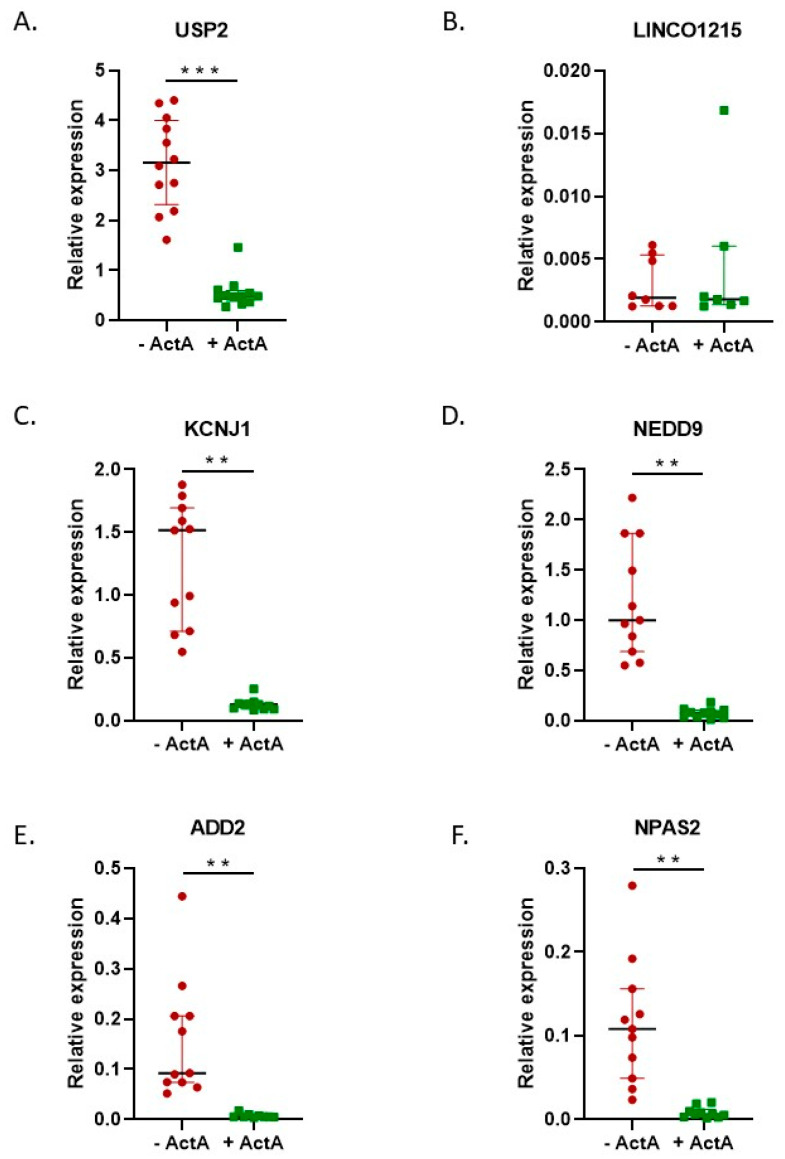
The qPCR confirmed the RNAseq data. The qPCR data for the most significantly differentially expressed genes depicted in the volcano plot. Data are presented as medians with the interquartile range. The qPCR data confirmed the RNA sequencing data. (**A**–**F**) qPCR of genes downregulated by Activin-A. −Activin-A, *n* = 11; +Activin-A, *n* = 11. Wilcoxon matched pairs signed rank test ** *p* < 0.01, *** *p* < 0.001.

**Table 1 ijms-24-06822-t001:** Donor demographics.

Donor	Sex	Age
Control 1	Female	27
FOP 1	Female	26
Control 2	Male	23
FOP 2	Male	19
Control 3	Female	24
FOP 3	Female	24
Control 4	Female	29
FOP 4	Female	30
Control 5	Female	25
FOP 5	Female	22
Control 6	Male	69
FOP 6	Male	67

**Table 2 ijms-24-06822-t002:** Primer sequences used for quantitative PCR.

Genes	Oligonucleotide Sequence, 5′–3′	Amplicon Length (bp)	Ensembl ID
*PBGD*	TGCAGTTTGAAATCATTGCTATGTC	84	ENSG00000256269
	AACAGGCTTTTCTCTCCAATCTTAGA		
*HTRA1*	ATCAAGGATGTGGATGAGAAAGC	61	ENSG00000166033
	GCTTGCCCTGGTGGTCAAT		
*RHOBTB1*	CAGTGTATGCTCCAAGTTCCGTAA	75	ENSG00000072422
	CGGTGCCGCTCGAAGTATT		
*USP2*	GGGAACACGTGCTTCATGAA	64	ENSG00000036672
	AATCTCTCAACTCCCGAGTGTTG		
*ANOS1*	CAGTGGCCCAGACCACAGA	59	ENSG00000011201
	CCATCGGCTGGGTCTTATGT		
*LINC01215*	AGAATGCACCTATTGGCTCACA	66	ENSG00000271856
	CTGCATTGTTATCATCACGACTTTC		
*HLA-DOA*	CAATCAAAGCCCATCTGGACAT	137	ENSG00000204252
	GTCCACGATGCAGATGAGGAT		
*GAL3ST4*	TCCTCTGTCACCACATGAGGTT	142	ENSG00000197093
	AAGGCTGATGAGGTGGATTTATAGTAG		
*KCNJ1*	TTCGGAAATGGGTCGTCACT	62	ENSG00000151704
	GGAGACTAGCCTTGCTCTTTGC		
*NEDD9*	CTGGATGGATGACTACGATTACGT	68	ENSG00000111859
	GCTCTTTCTGTTGCCTCTCAAAC		
*CIITA*	GCTCTACTCAGAACCCGACACA	63	ENSG00000179583
	TCACACAACAGCCTGCTGAAC		
*ADD2*	CCCAAGACCACGTGGATGA	75	ENSG00000075340
	TGGGTTTTCGATGCGAATC		
*OLFML3*	GGTGACAGACTGTGGCTACACAA	66	ENSG00000116774
	CCACCAAATCGCTTCAGAATC		
*PARM1*	CGTGGTGCTGCTGGTGTTT	61	ENSG00000169116
	TCCATAGGAGGAATGCCTGATT		
*NPAS2*	GGCAGCATCATCTATGTCTCTGA	67	ENSG00000170485
	CCATGACATCCGACGGTAAAT		
*OLR1*	CCAGCCTGATGAGAAGTCAAATG	72	ENSG00000173391
	AGGCACCACCATGGAGAGTAA		

*PBGD*, porphobilinogen deaminase; *HTRA1*, HtrA serine peptidase; *RHOBTB1*, Rho-related BTB domain containing 1; *USP2*, ubiquitin specific peptidase 2; *ANOS1*, anosmin-1; *LINC01215*, long intergenic non-protein coding RNA 1215; *HLA-DOA,* major histocompatibility complex, Class II, DO α; *GAL3ST4,* galactose-3-O-sulfotransferase 4; *KCNJ1*, potassium inwardly-rectifying channel subfamily J member; *NEDD9*, neural precursor cell expressed, developmentally downregulated 9; *CIITA*, Class II major histocompatibility complex transactivator; *ADD2*, Adducin 2; *OLFML3*, olfactomedin-like 3; *PARM1*, prostate androgen-regulated mucin-like protein 1; *NPAS2*, neuronal PAS domain 2; *OLR1*, oxidized low-density lipoprotein receptor 1.

## Data Availability

The RNAsequencing data are available under GEO accession number GSE228522.

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
