# Peer review of "Transcriptomic Differences Underlying the Activin-A Induced Large Osteoclast Formation in Both Healthy Control and Fibrodysplasia Ossificans Progressiva Osteoclasts"

_ijms, 2023, doi:10.3390/ijms24076822_

Round 1
Reviewer 1 Report (New Reviewer)
This study is a follow-up on a previously published study, where the authors observed that Activin-A induces the formation of larger human osteoclasts in culture. In the present study, a thorough transcriptomic analysis is performed in the search of molecular explanations for this observed effect.
The paper is well written, well presented, and relevant. Yet, I still have the following comments for improvement:
1) I find some inconsistencies in the formulations used with respect to the differentiation stage of 14 days. In lines 88-90 it is written as if this is at the early stages of fusion, while it in lines 258-260 is referred to as a "later stage". The authors argue that at 14 days this is where most fusion occurs. However, this is not supported by any data and it would be helpful for the reader to see some more details on what the stage at 14 days reflect. Therefore, the authors should support this statement with e.g. images from the different days of differentiation or TRAcP actity measurements from the conditioned media. It seems that these data were also not shown in the preceeding paper (Schoenmaker at al. 2020, Frintiers in Endocrinology, 11, article 501). This would improve the understanding of the findings and improve the impact of the paper.
2) In lines 404-406 the human donors are roughly presented as age and gender matched. It would be very helpful for the reader if a table is included that shows all known donor/patient demographics and the matched samples should be listed side by side so the study group is well presented.
3) Since there are both males and females included and since the age span is from 20 to 68 and since gender and age is known to affect osteoclast fusion in vitro it would be interesting to show if the gender or the age of the individuals makes the data cluster or affect the expression profile in any way? Is the Activin-A response altered with respect to gene expression by age or gender?
4) Line 404. This must be the wrong reference - I guess that this should be ref. 14? The authors should carefully check their references for any similar mistakes.
5) In Figures 7+8. I find it would be helpful to indicate the medians. As far as I can see they are not shown? Also the figure legend does not say what the error bars indicate. Based on the statistics that are used I guess it is the interquartile range?
6) A general comment to the writing style of the Results section is that the line between describing the results and discussing them is blurry in many places. I would advise to edit the Results section with this in mind and reduce the discussion-like elements because it distract the reader and you can loose a bit the focus on the results alone. Some of the discussion-like passages could very well be included in the Discussion section.
7) In line 243 it should say "genes" rather than "gens"
8) Please define the abbreviation "KEGG" (Kyoto Encyclopedia of Genes and Genomes). I do not think that this has been indicated in text at first mentioning.
Author Response
We thank the reviewer for the valuable comments. Please find our answers and the adjusted manuscript in the attached files.
Kind regards,
Ton Schoenmaker

Reviewer 2 Report (Previous Reviewer 1)
In the revised version of the MS, the authors have satisfactorily addressed my minor points of criscism. I have no further comments.
Author Response
We thank the reviewer for the valuable comments and for endorsing our manuscript.
Kind regards,
Ton Schoenmaker
Round 2
Reviewer 1 Report (New Reviewer)
The authors have made the possible and necessary improvements.
This manuscript is a resubmission of an earlier submission. The following is a list of the peer review reports and author responses from that submission.
Round 1
Reviewer 1 Report
In the present study, Schoenmaker et al. investigated the role of Activin-A in the development of osteoclasts, the most important bone-resorbing cells, at the level of gene expression. In fibrodysplasia ossificans progressiva (FOP), bone is formed in soft tissues such as tendons, muscles and ligaments. In addition to osteoblasts, osteoclasts showing a mutated activin receptor type 1 are involved in this rare disease. Cultured osteoclasts from both healthy controls and FOP patients were studied in the absence or presence of Activin-A. The data show that Activin-A has a marked effect on the upregulation of certain genes during osteoclast formation, in particular on such genes that trigger cell fusion. Interestingly, this effect is seen in both healthy controls and FOP patients with mutated ACVR1 receptor.
I think the study is well done and clearly written. I only have a few minor questions.
Results
How did the authors determine that the CD-14-positive monocytes differentiated into osteoclasts after 14 days of cultivation in the well plate?
Line (L) 143-145…..we found 1418 significantly differentially expressed genes in the presence of Activin-A, 566 of which were upregulated and 914 were downregulated. If I do the math, it's 566 + 914 = 1480.
Discussion
IL-6 is not mentioned as one of the "most upregulated genes". So why did the authors conclude that this molecule plays an important role in more or less all stages of the cell fusion process?
L 398 The myoblast is not a multinucleated cell. The myoblast is a mononuclear cell and several of these cells fuse to form a multinucleated skeletal muscle fibre.
L 409 - 426 Why did the authors mention hepatocytes? These cells usually have one nucleus and only occasionally two nuclei. They further discuss myoblast and monocyte fusion, but do not discuss hepatocytes further.
Author Response
Please see the attachement.

Reviewer 2 Report
The authors followed up their recent report that Activin-A induces fewer, but larger and more active osteoclasts in vitro in cultures of monocytes from healthy control and FOP patients, the latter known to have a disease-causing mutation in the ACVR1 receptor. To try to decipher the mechanisms underlying the formation of these large osteoclasts, they do RNAseq and standard bioinformatics analyses of Activin-A-induced osteoclastogenesis in the same kind of cultures from control and patient-derived CD14-positive monocytes at the late stage of the differentiation time course. They report that control and FOP cells respond very similarly to ActA and that 1480 genes were differentially expressed in response to Activin-A in both control and FOP cells, some of which genes are known to be associated with osteoclast differentiation or function, cell fusion or inflammation. They conclude that Activin-A has a substantial effect on gene expression during osteoclast formation and that this effect occurs in cells with either a WT or the FOP-causing mutated ACVR1 receptor. While the main goal of the work reported is interesting and the RNAseq data and analyses generally appear sound, I have several major concerns with the manuscript.
Specific comments
1. While the main goal of the work reported is worth pursuing, what is reported here is entirely descriptive and mechanistically has not significantly advanced our understanding of how ActA leads to fewer larger but more active osteoclasts, beyond that the authors reported in their earlier studies. The authors also do minimal validation of the changes they report, validation restricted to QPCR of some of the genes most highly up- or downregulated in response to ActA. On this, I could not see in Methods or Results a statement that the RNA used for validation was from samples independent of those used for the RNAseq; since the Fig. 7-8 legends refer to 11 treated and 11 untreated samples, it seems the QPCR was not done on independent samples. No functional analyses have been done. Thus, in both Results and Discussion, the authors speculate on what a wide variety of differentially expressed genes may be doing (based on existing knowledge) and on how ActA may elicit its effects in the presence of a mutated receptor (again, based on existing knowledge).
All that being said, the RNAseq data appear sound and can form a starting point for further work and analyses; in this regard, the data should be deposited in existing gene expression databases (I may have missed it, but couldn’t find a statement that the authors have deposited the data).
2. The manuscript is far too long and key messages can be reported much more clearly and concisely. For example:
Introduction – There is far too much background on what is known about ActA and osteoblasts in FOP, which has not been studied here (in any case, large sections of this text are repetitive of their 2020 paper). What the Introduction should be focused on is what they reported in the 2020 paper, what they learned from the analyses, including gene expression analysis they report there, and why they embarked on the RNAseq (some of this is in the 1st para of the Discussion but it should be in the Introduction with the rationale for the current studies). What would also be helpful in the Introduction/rationale and how the data are reported more generally, is why they focused RNAseq on a single time point - the late differentiation stage (d14). While it is obvious that this time point may ultimately provide additional insight into changes in late stage differentiation, fusion and resorptive activity, it won’t address the earlier stages for which they earlier reported some evidence for ActA-induced changes in known regulators such as M-CSF and DC-STAMP.
Methods and approach – Overall, the bioinformatics analyses are standard and can be presented in less detail in Methods and Results. Second, while the result is clear in fig. 1C to support the statement that ActA had similar effect on control and FOP cells, in Fig. 1B and throughout other figures, the legends refer to 11 samples with or without treatment; clarify whether these are control or FOP.
Results – Again, since the bioinformatics used to analyze the RNAseq data are standard, the KEGG and GO data can be summarized more succinctly, without the text repeating much of what is shown in the figures and explained in legends. Please correct: Volcano plots not Vulcano plots.
Discussion – This can be much shorter to summarize the findings without so much speculation on genes and pathways, none of which have been tested beyond what is already known. Second, the authors should be better integrating what they believe they have learned here versus what has been reported for osteoclastogenesis/osteoclasts in FOP bone by others and what they have reported earlier with normal versus FOP osteoclasts. Third, the limitations of the approach overall (e.g., the limitations of the cell culture approach, the single time point studied, the lack of functional analyses and lack of evidence for cause-effect for any of the changes seen) should be better highlighted and discussed.
Abstract and conclusions – As indicated above for the Introduction and Discussion, the Abstract could also be better focused. In addition, the rhetoric at end of the Abstract and Discussion related to treatment of FOP with ActA antibodies should be deleted; not only does it repeat almost verbatim what they wrote in their 2020 paper, nothing they have reported here advances our understanding of treatment with ActA antibodies.
Round 2
Reviewer 2 Report
As far as they go, the studies appear sound and the authors have made textual changes that improve the manuscript. However, I have essentially the same main concerns as previously. I agree with the authors that the results can serve as a starting point, but without additional studies, the work remains entirely descriptive, with no new understanding of FOP and minimal substantive or conclusive new understanding of Activin A effects on osteoclastogenesis.